# The Arctic Front and its variability in the Norwegian Sea

Roshin P. Raj[1], Sourav Chatterjee[2], Laurent Bertino[1], Antonio Turiel[3], Marcos Portabella[3]

[1]Nansen Environmental and Remote Sensing Center, Bjerknes Centre for Climate Research, Thormøhlens gate 47, Bergen, Norway

[2]National Centre for Polar and Ocean Research, Goa, India

[3]Barcelona Expert Centre, Institute of Marine Sciences (ICM-CSIC), Barcelona, Spain

*Correspondence to*: Roshin. P. Raj (roshin.raj@nersc.no)

**Abstract.** The Arctic Front (AF) in the Norwegian Sea is an important biologically productive region which is well-known for its large feeding schools of pelagic fish. A suite of satellite data, a regional coupled ocean-sea ice data assimilation system (the TOPAZ reanalysis) and atmospheric reanalysis data is used to investigate the variability in the lateral and vertical structure of the AF. A method, the so-called 'Singularity Analysis', is applied on the satellite and reanalysis data for 2D spatial analysis of the front, whereas for the vertical structure, a horizontal gradient method is used. We present new evidences of active air-sea interaction along the AF due to enhanced momentum mixing near the frontal region. The frontal structure of the AF is found to be most distinct near the Faroe Current in the southwest Norwegian Sea and along the Mohn Ridge. Coincidentally, these are the two locations along the AF where the air-sea interactions are most intense. This study investigates in particular the frontal structure and its variability along the Mohn Ridge. The seasonal variability in the strength of the AF is found to be limited to the surface. The study also provides new insights on the influence of the three dominant modes of the Norwegian Sea atmospheric circulation on the AF along the Mohn Ridge. The analyses show a weakened AF during the negative phase of the North Atlantic Oscillation (NAO-), even though the geographical location of the front does not vary. The weakening of AF during NAO- is attributed to the variability in the strength of the Norwegian Atlantic Front Current over the Mohn Ridge associated with the changes in the wind field.

# 1 Introduction

Ocean fronts are boundaries between distinct water masses with large gradient in temperature or salinity (e.g., Bakun, 1996; D'Asaro et al., 2011). The Arctic Front (AF; Swift and Aagaard, 1981; Piechura and Walczowski, 1995) is one of the most prominent ocean fronts in the Norwegian Sea. Similar to its counterparts in the world ocean, the AF is an important biologically productive region also known for its large feeding schools of pelagic fish (e.g. Holst et al., 2004; Blindheim and Rey, 2004; Melle et al., 2004). On its influence on higher trophic levels, it is important to note that the Jan Mayen Island located near the AF is an important breeding region inhabited by large colonies of seabirds (Norway Ministry of Environment, 2008-2009).

The AF in the Norwegian Sea extends from the Iceland-Faroe Plateau to the Mohn-Knipowich Ridge (Nilsen and Nilsen, 2007) and is associated with the interaction of the warm and saline Atlantic Water and the cold and fresher Arctic Water (Blindheim and Ådlandsvik, 1995). As shown in Fig. 1, the Atlantic Water is carried into the Norwegian Sea via the Norwegian Atlantic Current (e.g., Orvik et al., 2001; Raj et al., 2016), which is a two-branch current system, with an eastern branch following the shelf edge as a barotropic slope current, and a western branch following the western rim of the Norwegian Sea as a topographically guided front current (Poulain et al., 1996; Orvik and Niiler, 2002; Skagseth and Orvik, 2002; Orvik and Skagseth, 2003). These two branches are known as the Norwegian Atlantic Slope Current (NwASC; Skagseth and Orvik, 2002) and the Norwegian Atlantic Front Current (NwAFC; Mork and Skagseth, 2010) respectively. On its poleward journey, the NwASC bifurcates, one part flowing into the Barents Sea and the other continuing towards the Fram Strait as the core of the West Spitsbergen Current (Helland-Hansen and Nansen, 1909, Aagaard et al., 1985). The NwAFC on its way to the north encounters three deep currents (Fig. 1), one over the Mohn Ridge flows in the opposite direction (Orvik, 2004). The NwAFC continues poleward, topographically guided by the Mohn Ridge and the Knipovich Ridge, as the western branch of the West Spitsbergen Current (WSC; Walczowski and Piechura, 2006). The Atlantic Water carried by the western branch of the West Spitsbergen Current recirculates mostly within the Nordic Seas, as the Return Atlantic Water (RAW; Eldevik et al., 2009). This AW mixes with the Polar Waters transported south from the Arctic by the East Greenland Current and forms the hydrographically distinct Arctic Water (Blindheim and Østerhus, 2005). The further interaction of the Arctic Water with warm and saline Atlantic Water results in the AF (Swift, 1986). The location of the AF follows the topography and coincides with the location where Arctic Water meets the Atlantic Water.

The AF has been subject of many studies (e.g., Piechura and Walczowski, 1995; Nilsen and Nilsen, 2007). However, the impact of the large-scale atmospheric forcing on the spatial (lateral) and vertical variability of the front has not yet been described. Our study aims to examine the structure (lateral and vertical) of the AF using satellite and reanalysis data, on climatological-mean and seasonal-mean timescales (1991-2015). The near permanent structure of the AF justifies its investigation on long timescales. In addition, the study aims to examine the influence of the three dominant modes of the large-scale atmospheric forcing in the Norwegian Sea, i.e., the North Atlantic Oscillation (NAO), the East Atlantic Pattern (EAP) and the Scandinavian Pattern (SCAN), on the variability of the AF. NAO is the most dominant atmospheric mode in the North

Atlantic and Nordic Seas (Fig. 2). Even though the impact of NAO on the NwASC is well-known, its impact on the NwAFC and on the AF is not clearly documented. Furthermore, it has been reported that the location of the centers of the NAO dipole can be affected through the interplay with EAP and SCAN teleconnection patterns (e.g., Moore et al., 2012; Chafik et al., 2017). The independent effect of EAP and SCAN on the AF has also been not investigated yet. We perform composite analysis (see Section 2) on the monthly ocean and atmospheric reanalysis data in order to capture the variability and thus to delineate the impact of the 3 main atmospheric modes of the Norwegian Sea on the AF. In Section 2, we describe the different datasets and methods used in this study. The results are presented in Section 3 and are discussed and summarized in Section 4.

## 2 Data

### 2.1 TOPAZ Reanalysis data

TOPAZ is a coupled ocean and sea ice data assimilation system for the North Atlantic and the Arctic that is based on the Hybrid Coordinate Ocean Model (HYCOM) and the Ensemble Kalman Filter data assimilation, the results of its 4th version (TOPAZ4) have been extensively validated (e.g., Lien et al., 2016; Xie et al., 2016; Chatterjee et al., 2018). The ocean model in TOPAZ4 is an eddy permitting model with 28 hybrid z-isopycnal layers at a horizontal resolution of 12 to 16 km in the Nordic Seas and the Arctic. TOPAZ4 represents the Arctic component of the Copernicus Marine Environment Monitoring Service (CMEMS) and is forced by the European Centre for Medium-Range Weather Forecasts (ECMWF) Reanalysis (ERA Interim) and assimilates measurements including along-track altimetry data, sea surface temperatures, sea ice concentrations and sea ice drift from satellites along with in-situ temperature and salinity profiles. The monthly TOPAZ4 results used in this study for the time period 1991-2015 have been obtained via the Copernicus marine services (marine.copernicus.eu).

### 2.2 Satellite and atmospheric reanalysis data

Advanced Microwave Scanning Radiometer (AMSRE; 2002-2011; 0.25° grid) SST and QuikSCat winds (1999-2009; 0.25° grid) are the satellite datasets used in this study. These monthly gridded datasets are obtained from http://apdrc.soest.hawaii.edu. For the analysis using satellite data, this study only uses the data from the overlapping time-period of AMSRE and QuikSCat, i.e., 8 years (2002-2009). In addition, monthly outputs (winds and mean sea level pressure (MSLP); 0.25° grid) from the ECMWF ERA Interim (Dee et al., 2011), which also assimilates the above-mentioned satellite datasets during the above-mentioned coincident periods, is used in this study over a longer time period, i.e., 1991-2015.

## 3 Methods

### 3.1 EOF analysis

The monthly detrended and deseasoned ERA Interim MSLP data is used to calculate the three leading empirical orthogonal functions (EOFs) of atmospheric variability in the North Atlantic (80ºW to 50ºE and 30ºN to 80ºN), which is based on a singular value decomposition, according to the method described by Hannachi et al. (2007). The data are weighted by the

cosine of its latitude at every grid point of the study region to account for decreasing grid sizes towards the pole. The three leading modes of atmospheric variability in the North Atlantic are the North Atlantic Oscillation (NAO), the East Atlantic Pattern (EAP), and the Scandinavian Pattern (SCAN). These three EOFs explain respectively 34%, 18% and 16% of the MSLP variability over the north Atlantic (Fig. 2). Eventhough EAP and SCAN individually explains less than 20% of MSLP variability (EAP: 18%; SCAN: 16%), several studies found that its effect on the Nordic Seas cannot be neglected (Comas-Bru and McDermott, 2013; Chafik et al., 2017). The total variance (68%) explained by the three modes estimated in our study is comparable to those computed in earlier studies using the NCEP data (60%; Chafik et al., 2017). Standardized principle components of the three modes are used as indices for NAO, EAP and SCAN respectively (Fig S1).

## 3.2 Composite analysis

For composite analysis, the positive/negative phase of NAO (NAO+/NAO-) are categorized as those months with NAO index values (Section 3.1) above/below one standard deviation calculated over the entire period from 1991 to 2015. Similarly, the positive/negative phase of EAP and SCAN are estimated from their respective time series (Fig S1). Significance of difference in the composite mean and climatological mean is estimated using the two-sample t-test for equal means (Snedecor and Cochran, 1989).

## 3.3 Singularity analysis

There are several methods used in previous studies to identify ocean fronts from satellite data (e.g., Cayula and Cornillon, 1992, 1995; Garcia-Olivares et al., 2007; Turiel et al., 2008). One of them is the Singularity analysis, which has been established as a powerful tool to detect frontal structures and streamlines from satellite images (Turiel et al., 2008, 2009; Portabella et al., 2012; Lin et al., 2014; Umbert et al., 2015). Around three and half decades ago, Mallat and Huang (1992) introduced Singularity analysis of scalar variables in the context of wavelet analysis. The Singularity analysis aims to obtain a dimensionless measure known as the singularity exponent at each point, which represents the degree of irregularity at that location. Singularity exponents are dimensionless and can be derived from any scalar quantity, for e.g., SST (Turiel et al., 2009) and wind components (Portabella et al., 2012; Lin et al., 2014). Singularity exponent is a continuous extension of classical concepts such as continuity or differentiability. The main difference between the maximum gradient method and singularity exponents is that singularity exponents are normalized so the absolute value of the gradient is irrelevant, what is important is the degree of correlation between nearby gradients, and singularity exponents are the dimensionless measures of that correlation. Hence the results from the singularity analysis of different scalars variables (for e.g., SST and windspeed) can be directly compared. Furthermore, the Singularity analysis does not require knowledge of the velocity field, because it is a Eulerian method exploiting the scaling properties of the spatial correlations of the gradients of a given scalar field. Thus, the Singularity analysis has an advantage over the use of Lyapunov exponents (Garcia-Olivares et al., 2007), another widely used methodology which requires the velocity field to be known.

Our study uses Singularity analysis to detect the AF in the Norwegian Sea from satellite data (SST and wind speed) as well as from ocean reanalysis data (temperature). Singularity exponents of the above scalar variables have been estimated using an online service provided by the Barcelona Expert Center (http://bec.icm.csic.es/CP34GUIWeb). The online toolbox first evaluates the modulus of the gradient of the scalar at each point in the specified domain, i.e., the Norwegian Sea in our study. A wavelet projection is applied on the modulus of the gradient of the variable as it has been shown to be useful to remove long-range correlations and thus reveal the actual local behavior of the function (Turiel et al., 2008). Then a log-log regression of the wavelet projection versus the resolution scale is used to determine the scaling properties of the gradient correlations, and the slope of such a regression is precisely the singularity exponent. Therefore, Singularity analysis assigns a singularity exponent to each point in an image. The singularity exponent value provides information about the local regularity (if positive) or irregularity (if negative) of the signal at the given point, being in fact an extension of the concept of differentiability (Turiel et al., 2008). Due to its connection with the theory of turbulent flows, it has been shown that the singularity exponents obtained from any ocean scalar delineate the streamlines of the flow. Details of the method is provided in the Appendix Section.

## 4 Results and Discussion

### 4.1 Surface structure of the Arctic front

The spatial distribution of the climatological (2002-2009) SST of the Norwegian Sea, shown in Fig. 3a illustrates a typical example of the distribution of surface waters in the Norwegian Sea. The figure shows that the warmest waters are confined to the eastern Norwegian Sea, while the coldest waters are found in the Greenland Basin west of the Mohn Ridge. From a closer inspection, the comparatively large temperature gradient associated with the AF at the Mohn Ridge (Piechura and Walczowski, 1995) can be identified. This large temperature gradient over the Mohn Ridge is associated with the warm Atlantic Water in the Lofoten Basin and the colder Arctic Waters in the Greenland Basin. In addition to SST, we use remotely sensed winds to identify the ocean fronts in the Norwegian Sea, as has been done for parts of the global ocean (e.g., Song et al., 2006). Fig. 3b shows the spatial distribution of the climatology (2002-2009) of wind speed in the Norwegian Sea. Note that the choice of time-averaged fields assists in avoiding the possible contamination of the wind field by the rapid evolution of weather patterns (Chelton et al., 2004). Comparison of Fig. 3a and 3b reveals stronger/weak winds over warm/cold waters. Even though shown for the first time in the Norwegian Sea, the increase/decrease in surface wind speed when it blows from cold/warm to warm/cold water is well-known and the physical mechanism has been extensively studied (e.g., Friehe et al., 1991; Chelton et al., 2004; Song et al., 2006). The response of the surface wind to the temperature fronts has been studied on different timescales (monthly, Chelton et al., 2001; seasonal, O'Neill et al., 2003; and climatological, Chelton et al., 2004) especially in the western boundary currents. Stronger surface winds over warmer waters are due to efficient turbulent convection that transfers momentum down to the surface (Wallace et al., 1989). This process known as momentum mixing mechanism destabilizes air over warm water, and the increased turbulent mixing of momentum accelerates near-surface winds. Conversely, cold SST

suppresses the momentum mixing, decouples the near-surface wind from wind aloft, and decreases the near surface wind. Another interesting feature seen in the wind field is the cold tongue of Arctic Water in the southern Norwegian Sea (~63°N, 8°W to 4°W; location 2 in Fig. 4a). This cold tongue is caused by mixing of the water mass carried by the North Icelandic Irminger Current into the surrounding waters, mostly in the east Icelandic Current, which flows into the southwest Norwegian
Sea, where it forms a cold tongue of Arctic character (Helland-Hansen and Nansen, 1909; Blindheim and Malmberg, 2005). Note that the main focus of the present study is to study the mean and variability of the AF. The SST-wind interaction over the AF revealed in this study warrants a detailed analysis and needs to be mathematically formulated, and hence is out-of-scope of this study.

       Singularity exponents estimated from these mean satellite SST and wind speed fields, as shown in Fig. 4, provide a
clearer and consistent picture of the AF in the Norwegian Sea compared to the original scalar fields. The AF in the Norwegian Sea are characterised by negative singularity exponents. Previous studies using singularity exponent analysis also found similar results in frontal regions (e.g., Fig. 5 in Turiel et al., 2008). Regarding the SST, regions with negative singularity exponents correspond to regions with higher irregularity, i.e., to grid points with stronger gradient variations (see Figs. 3a and 4a for SST) than the neighbouring grid points (sharp transition). On the other hand, regions with positive singularity exponents are those
where the gradient variations are weak. The frontal region at the Mohn Ridge (location 1 in Fig. 4a) and that associated with the cold tongue and the Faroe Current in the southwest Norwegian Sea (location 2) are the locations where the AF is most prominent. The imprint of East Icelandic Current can also be seen in Fig. 4a (location 3). The AF extending northeast from the cold tongue region is locked between the 1800 m and 2400 m depth contour (location 4). The signature of the AF at location 4 is not very prominent compared to locations 1 and 2. Further north over the Vøring Plateau (location 5), the continuation of
the AF is not captured in our analysis. Note that this is also a region of intense mesoscale eddy activity. The tracks of mesoscale eddies in the region obtained from Raj et al. (2016; updated time series till 2016) is shown in Fig. S2. The figure shows dominance of mesoscale eddies across the AF at locations 4 and 5, more prominent at location 5. Eddies are known to contribute significantly to the total oceanic heat and salt transport by advective trapping (Dong et al., 2014), stirring and mixing (Morrow and Le Traon, 2012). Most importantly they can play an import role in the cross-frontal transport (Dufour et al.,
2015). Eddy-induced mixing and cross-frontal transport can reduce the sharp distinction of the water masses at the AF in location 5 compared to the other four. This can be a reason for the absence of the AF signature in the singularity exponents map. Quantifying the impact of mesoscale eddies on the AF needs a dedicated effort and is out of the scope of our study.

       All the above SST-related features are also portrayed in the map of singularity exponents of the wind speed (Fig. 4b), estimated from the mean satellite wind speed map (shown in Fig. 3b). Regions with the sharpest changes in wind speed are
consistent with those found in the SST map (Fig. 4a). In other words, regions where SST is found to experience sharp changes correspond well to those with large changes in wind speed, thereby confirming the role of strong momentum mixing along the frontal regions. The analysis is repeated using other remote sensing datasets (WindSAT SST, AVHRR SST, WindSAT wind speed and AMSRE wind speed) in order to examine whether the ocean fronts in the Norwegian Sea can be retrieved irrespective

of the satellite data and sensor used. Singularity Analysis on these satellite SST and wind speed datasets also illustrates similar AF signatures to those seen in Fig. 4 (not shown). Hence it can be stated that the satellite-derived wind speed and SST data reveal the frontal structure of the AF, irrespective of the sensor used. Thus, it is shown that Singularity Analysis is able to portray a much clearer representation of the AF, compared to the original scalar fields.

## 4.2 Subsurface structure of the Arctic Front

Next, we assess the performance of the TOPAZ4 reanalysis results in reproducing the horizontal structure of the AF in the Norwegian Sea, using a longer period (1991-2015) than that of the satellite data (2002-2009). The surface signature of the AF estimated from TOPAZ4 SST output (Fig. 5a) agrees with that of the satellite SST data (Fig.4a). As mentioned in Section 2.1, TOPAZ4 reanalysis assimilates measurements including along-track altimetry data, sea surface temperatures, sea ice concentrations and sea ice drift from satellites along with in-situ temperature and salinity profiles. Hence, the similarities in Fig.4a and 5a are not surprising, although not warranted due to the residual errors of assimilation. Nevertheless, the figures confirm that TOPAZ4 reanalysis data is able to reproduce the AF signature and indicates that the TOPAZ4 results can be used to study the subsurface part of the AF. Further analysis show that the horizontal structure of the AF in deeper ocean is similar to the surface: most distinct along the Mohn Ridge and near the Faroe Current in the southwest Norwegian Sea (as shown by the TOPAZ4 potential temperature singularity exponent maps in Figs. 5b and 5c).

Next, in order to examine the seasonal variability of the AF in the Norwegian Sea, the Singularity Analysis is applied to the seasonal climatologies of SST and potential temperature at 200 m (Fig. 6). The surface signature of AF is found to be stronger during winter (DJF; Fig. 6a), while the frontal signature is less evident during summer (JJA; Fig. 6b). These results agree with the analysis done using seasonal satellite SST data (not shown). Note that the analysis of the subsurface temperature fields (Figs. 6c and 6d) does not reproduce the seasonal variability of the surface frontal structure (Figs. 6a and 6b). Thus, it can be concluded that the seasonal variability of the frontal structure is limited to the surface and not found in the subsurface. This is likely to be associated with the surface heating of the ocean during summer that in turn reduces the surface horizontal temperature gradient, evidence of which is found as a thin layer of (upper 25 m) low temperature gradient near the surface during summer (Fig. 7c), which is not present in winter (Fig. 7b), that disconnects the subsurface signatures of the front from the surface.

The TOPAZ4 reanalysis, which successfully replicates the lateral variability of the AF (Fig. 5-6), is further used to investigate the vertical structure of the AF and its variability. The vertical structure of the AF is analyzed using a simple horizontal gradient method. The main reason for using a simple horizontal gradient method to estimate the strength of the front is to compare our results with previous studies (e.g., Piechura and Walczowski, 1995; Lobb et al., 2003). Furthermore, for this analysis the focus area is limited to a single section (shown in Fig. 4a) taken across the Mohn Ridge, where a strong frontal signature is found in the singularity exponents maps (Fig. 4-6). Analysing the variability of the AF over the Mohn Ridge is also important due to its proximity to the Jan Mayen Island, which is well-known to be an important breeding region inhabited by large colonies of seabirds. Although higher up in the food chain, the variability of the AF may have an influence on the

birds through the impact on biology and fisheries of the region. A similar section was used by Piechura and Walczowski (1995), to study the AF across the Mohn Ridge using CTD data during the time period 1987-1993. Fig. 7a-c shows respectively the mean, winter and summer climatology of potential temperature along the vertical cross-section at the Mohn Ridge (location shown in Fig 4a). To the east (left side of the figure), the figure shows the warm Atlantic Water residing in the Lofoten Basin,

while the cold waters reside in the Greenland Sea (right side of the figure). Note that these results agree with those reported by Piechura and Walczowski (1995). The temperature gradient along the vertical section delimits the AF at the Mohn Ridge (Fig. 7d). The location of the core of the front along the section across the Mohn Ridge is marked as red in Fig. 4a. At the location of the AF, a temperature change of roughly 2 degrees can be noted from Fig. 7a. The potential temperature gradient (0.04 degree/km) across the core of the AF is comparable with those reported at other high-latitude frontal regions (Lobb et

al., 2003). The AF extends from the surface down to 600 m depth with the core stretching over depths from 50 m to 400 m, as indicated by the strong potential temperature horizontal gradient areas (dark blue contours) in Figs. 7d-f. The frontal structure is well connected to the surface during winter (Figs. 7b and 7e), while in summer, the stratified layer of warm water at the surface (upper 25 m) disconnects the AF deeper layers from the surface (Figs. 7c and 7f). The figures also show that there is no seasonal variability in the location of the core of the AF over the Mohn Ridge.

**4.3 Impact of large-scale atmospheric forcing on the Arctic front**

The impact of the large-scale atmospheric forcing in the Norwegian Sea (Fig. 2) on the variability of the AF over the Mohn Ridge is investigated next using composite analysis on monthly TOPAZ4 data. For this, composite maps (Section 3.2) of the temperature across the Mohn Ridge during the positive and negative phases of NAO, EAP and SCAN are produced and the gradient in mean temperature is estimated. The analysis shows that the location of the core of the AF along the Mohn Ridge

is not influenced by the large-scale atmospheric variability (Fig. 8, Fig. S3). Here, the AF is defined as the region with the maximum in temperature gradient, a classical definition used to distinguish between two distinct water masses, in this case the Atlantic Water and the Arctic Waters. The non-variability in the location of the position of the AF further indicates that the location of the two water masses does not alter. However, Fig. 8b shows a weakening of the AF during the negative phase of NAO (NAO-). The weakening of the AF during NAO- is prominent in the upper 450 m depth (Fig. 8b). Compared to NAO,

the 2nd (EAP) and 3rd (SCAN) modes of atmospheric variability in the Norwegian Sea do not show a major impact on the AF over the Mohn Ridge (Fig. S3). Further analysis thus only focuses on the two opposite phases of NAO. The atmospheric circulation associated with the positive and negative phases of NAO estimated using the ERA Interim surface winds is shown in Fig. 9. During NAO+, the composite map shows a distinct anomalous cyclonic pattern, centered on the Barents Sea Opening (Fig. 9a). In contrast, during NAO-, a weakened cyclonic circulation is found (shown as anomalous anticyclonic circulation)

centered near the Mohn Ridge (Fig. 9b).

Next, we investigate the link between the atmospheric circulation during NAO- (Fig. 9b) and the weakening of the AF over the Mohn Ridge (Fig. 8b). We hypothesise that the impact of NAO on ocean currents of the region may influence the AF. Previous studies have reported the impact of the atmospheric variability on the ocean currents in the Norwegian Sea (e.g.,

Skagseth et al., 2008; Raj et al., 2018). Over the Mohn Ridge the NwAFC transports warm AW poleward. Note that although the vertical section over the Mohn Ridge is aligned almost perpendicular to the AF, the maximum velocity component of the NwAFC and the gyre circulation are not necessarily perpendicular to the section, and hence cross-section velocities may not give a correct representation of the currents. Instead, the speed along the section can give a better representation of the currents, especially since the direction of the mean currents over the region does not change. Our results (Fig. 10) show that the core of the NwAFC over the Mohn Ridge, is narrower and strengthened, especially at depths during NAO-, while during NAO+ it is broader and comparatively weaker in the subsurface (Fig. 10). The narrowing of the NwAFC associated with its strengthening during NAO- has resulted a shift of its core by about 20-40 km away from the AF (Fig. 10b). Also, the strength of the NwAFC in the region where it gets narrower (AF indicated by dashed line in Fig 10b) is also significantly different (shown as stripped in Fig. 10b) from the climatological mean.

While majority of previous studies reported the positive relation of NAO with the speed of the NwASC (e.g., Skagseth and Orvik, 2002), our results show that the NwAFC behaves in an opposite way. One of the possible mechanisms by which a weakening in the cyclonic atmospheric circulation during NAO- (Fig. 9b) results in the strengthening of the NwAFC (Fig. 10b) is via its influence on the cyclonic gyre circulation of the Lofoten Basin. Near the Mohn Ridge, the cyclonic gyre circulation of the Lofoten Basin is counter-propagating in comparison to the northward flow of the NwAFC (see Fig.1). It has been argued that this counter-propagating part of the cyclonic gyre acts as a hindrance to the northward flow of the NwAFC, which in turn results in the weakening of the NwAFC (Orvik, 2004). In his study using a reduced gravity model, Orvik (2004) showed that a weakening of the gyre circulation will result in the strengthening of the NwAFC as the hindrance is reduced. Regarding the link between NAO- and Lofoten Basin gyre circulation, we hypothesise that a weakening of the cyclonic atmospheric circulation during NAO- can result in the weakening of the Lofoten Basin gyre circulation. The evidence of this atmospheric-ocean interaction is shown in Fig. 11. The figure shows the composite of the monthly barotropic stream function (BSFD) data (estimated from depth-averaged currents) from the TOPAZ4 reanalysis, and exhibits the gyre circulations in the Lofoten Basin, the Greenland Sea and the Norwegian Basin. The negative values seen in Fig. 11a during NAO+ indicate stronger cyclonic circulation. According to the figure, the cyclonic Lofoten Basin gyre circulation during NAO+ is found to be intensified. On the contrary, a weakening of the Lofoten Basin gyre circulation is found during NAO- (Fig 11b; positive values indicate a weaker cyclonic circulation). This weakening in the gyre circulation of the Lofoten Basin during NAO- (Fig. 11b) can result in the strengthening of NwAFC (Fig. 10b) as there is less hindrance to the northward flow.

The discussion above explains the link between the variability in the atmospheric circulation during NAO- (Fig. 9b) and the strengthening and narrowing of the NwAFC over the Mohn Ridge (Fig. 10b). Associated with the strengthening and narrowing of the NwAFC during NAO-, our study also identified a shift (20-40 km) in the core of the NwAFC away from the core of the AF. A shift in the core of the NwAFC from the topographically trapped AF can result in the decrease in temperature gradient across the front, in turn leading to the weakening of the front (Fig. 8b). A schematic illustration of the different processes connecting NAO- to a weakened AF as explained above is shown in Fig. 12. However, note that the physical

mechanism explained in Fig. 12 may not be applicable to the variability of AF at location L2 in Fig. 4a, another location where the AF is found to be most prominent (Fig. 4-6). This is due to the fact that the circulation pattern over the Mohn Ridge is different compared to that at L2. I.e., while the gyre circulation in the Lofoten Basin is opposite to the northward flow of the NwAFC over Mohn Ridge, the flow of the NwAFC at location L2 follows the direction of the gyre circulation of the region (Fig. 1). The impact of NAO on the AF at location L2 needs to be analysed in detail since anomalous winds were found over the region during opposite phases of NAO (Fig. 9). We recommend a separate study on the topic since it is out of the scope of this study.

## 5 Summary

Satellite-derived SST and wind speed data are good upper level proxies for the hydrographic signature of semi-permanent ocean fronts across the Norwegian Sea. They reveal the frontal structure of the AF, irrespective of the sensor used. The AF is found to be strongest over the Mohn Ridge and near the Iceland Faroe Plateau. The study shows evidences of active air-sea interaction along the AF. Compared to the original scalar fields (SST and wind speed), a more precise picture of the AF in the Norwegian Sea is obtained from their corresponding singularity exponents. The AF at the Mohn Ridge extends from the surface down to 600 m depth. The seasonal variability of the AF is limited to the surface and associated with the surface heating of the ocean during summer that reduces the horizontal surface temperature gradient. There is no seasonal variability in the location of the AF core over the Mohn Ridge. For the first time, the role of the three dominant atmospheric modes in the Norwegian Sea on the variability of the AF is investigated. Out of the three, only the NAO influences the AF along the Mohn Ridge. The study reveals a profound weakening of the AF over the Mohn Ridge during NAO-, which is further associated with the variability of the prevailing atmospheric circulation. A weakening of the cyclonic atmospheric circulation during NAO- weakens the deep gyre circulation of the Lofoten Basin which in turn results in a strengthened and narrower NwAFC in opposite direction. The narrowing of the NwAFC causes its core to shift eastwards (20-40 km) from the core of the topographically trapped AF, which leads to a decreased temperature gradient across the front, and in turn to the weakening of the front.

## Acknowledgements

This research is funded by the Copernicus Arctic MFC services. We also acknowledge the CPU provided by the Norwegian supercomputing project Sigma2 (nn2993k) and the storage space (ns2993k). Author Sourav thanks Nansen scientific Society for the support. We also thank Dr. Richard. E. Danielson for his helpful suggestions.

## Data/Code Availability

All datasets used in this study are freely available. TOPAZ model data are available via the CMEMS portal (http://marine.copernicus.eu). The satellite datasets used here are available at http://apdrc.soest.hawaii.edu. ERA interim data

is available at https://apps.ecmwf.int/datasets/data/interim-full-moda/levtype=sfc/. The online toolbox used for the estimation of singularity exponents of the satellite and TOPAZ data is available at http://bec.icm.csic.es/CP34GUIWeb (free registration).

**Author Contribution**

RR initiated the collaboration and designed the outline of the paper. SC and RR were instrumental in the data analysis. RR, SC, LB, AT and MP contributed to the interpretation of results. RR led the writing of the paper with significant contributions from all co-authors, at all stages of the paper.

**Competing interests.**

The authors declare that they have no conflict of interest.

**Appendix A: Singularity analysis**

In 1941, Kolmogorov (Kolmogorov, 1941) introduced the concept of scale invariant dissipations. Turbulence is characterized by an infinite amount of degrees of freedom and thus it is impossible to fully characterize what happens at each element of the fluid. However, the large number of degrees of freedom allowed for a statistical approach, and thus turbulent flows could be characterized by effective quantities. What Kolmogorov proposed was that there is a simple relation between the energy dissipation at a scale L and the energy dissipation at a smaller scale r, as:

$$\frac{\varepsilon_r}{\varepsilon_L} = \left(\frac{r}{L}\right)^\beta \tag{A1}$$

Where $\varepsilon_r$ is the dissipation at the scale r and $\varepsilon_L$ is the dissipation at the larger scale L. This relation was called "normal scaling" as it implies a simple distribution of energy across scales; in fact, it is sometimes referred as linear scaling because equation (A1) implies that plot of the logarithm of $\varepsilon_r$ versus the logarithm of r should be a straight line of slope β. It was soon realized that "normal scaling" was not frequent at all. On the contrary, the plots of the logarithm of the energy dissipation at scale r versus the logarithm of r generally showed a concave curve, the so-called "anomalous scaling", thus revealing that there was a much richer statistical interplay between the scales. Equation (A1) can be generalized in a stochastic way to account for that fact, namely:

$$\varepsilon_r \sim \eta_{\frac{r}{L}} \, \varepsilon_L \tag{A2}$$

Where $\eta_{\frac{r}{L}}$ is a random stochastic variable which only depends on the ratio of scales r and L, and the symbol ~ means that both sides have identical statistical distribution. The variable $\eta_{\frac{r}{L}}$ is called the cascade variable, because it defines a cascade relation (its distribution for a change in scale L to r must be the same if we change first from scale L to any intermediate scale l and then from l to r).

In 1985, Parisi and Frisch (Parisi and Frisch, 1985) first introduced the concept of multifractal to explain that in fact the properties of cascade variables could be understood if we supposed that there is a hierarchy of fractal components $F_h$, each one associated to a particular local dissipation exponent h, and such that the fractal dimensions of each fractal component

could be related with the distribution of the cascade variables. There has been an extensive literature by Turiel and collaborators explaining the connection between this microscopic behavior (the local exponents h) and the global statistical properties, what led to the introduction of singularity analysis. By means of singularity analysis we want to determine the local properties of scaling of the energy dissipation or any other scalar variable in a turbulent flow. To evaluate singularity analysis, we must proceed by applying continuous wavelet transforms to the variable under study (let us call it θ), namely

$$T_\psi |\nabla\theta|(\vec{x}, r) \equiv \int d\vec{y} \; \frac{1}{r^2} \; \Psi\left(\frac{\vec{y}-\vec{x}}{r}\right) |\nabla\theta|(\vec{y}) \tag{A3}$$

Where r is the scale of observation and x is the point at which the wavelet is been applied. We apply the wavelet projection on the modulus of the gradient of the variable because it has been shown to be useful to remove long-range correlations and thus reveal the actual local behavior of the function (Turiel et al., 2008). For turbulent fluids, according to Parisi and Frisch (1985) the cascade relation translates into the wavelet projection as:

$$T_\psi |\nabla\theta|(\vec{x}, r) \propto r^{h(\vec{x})} \tag{A4}$$

Therefore, a log-log regression of the wavelet projection versus the logarithm of the scale allows to calculate the singularity exponent h(x), which would be the slope of that regression. Something important to notice about singularity exponents is that they are unitless: they are dimensionless measurements of the local correlation of the gradient of the variable θ at the point x with the same variable at the neighbouring points. Therefore, changes in the amplitude of the variable are irrelevant, as they just change the proportionality constant in (A4) that is not accounted in the log-log regression because it only contributes to the intercept (and h(x) is the slope of that regression). In fact, local changes in amplitude that change slowly enough across the area do not change the singularity exponents either. Singularity analysis is then more robust than other approaches as Wavelet Modulus Transform Maxima, because it is not affected by those artifacts, and does not imply any particular condition on the geometry of the singularity fronts (WTMM require maxima points to be isolated); in addition, singularity analysis is in a close connection with the turbulent properties of the flow. The choice of the particular wavelet allows for a better discrimination of the singularity exponents at the resolution scale; in this paper we have chosen the same as in Pont et al. (2013).

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

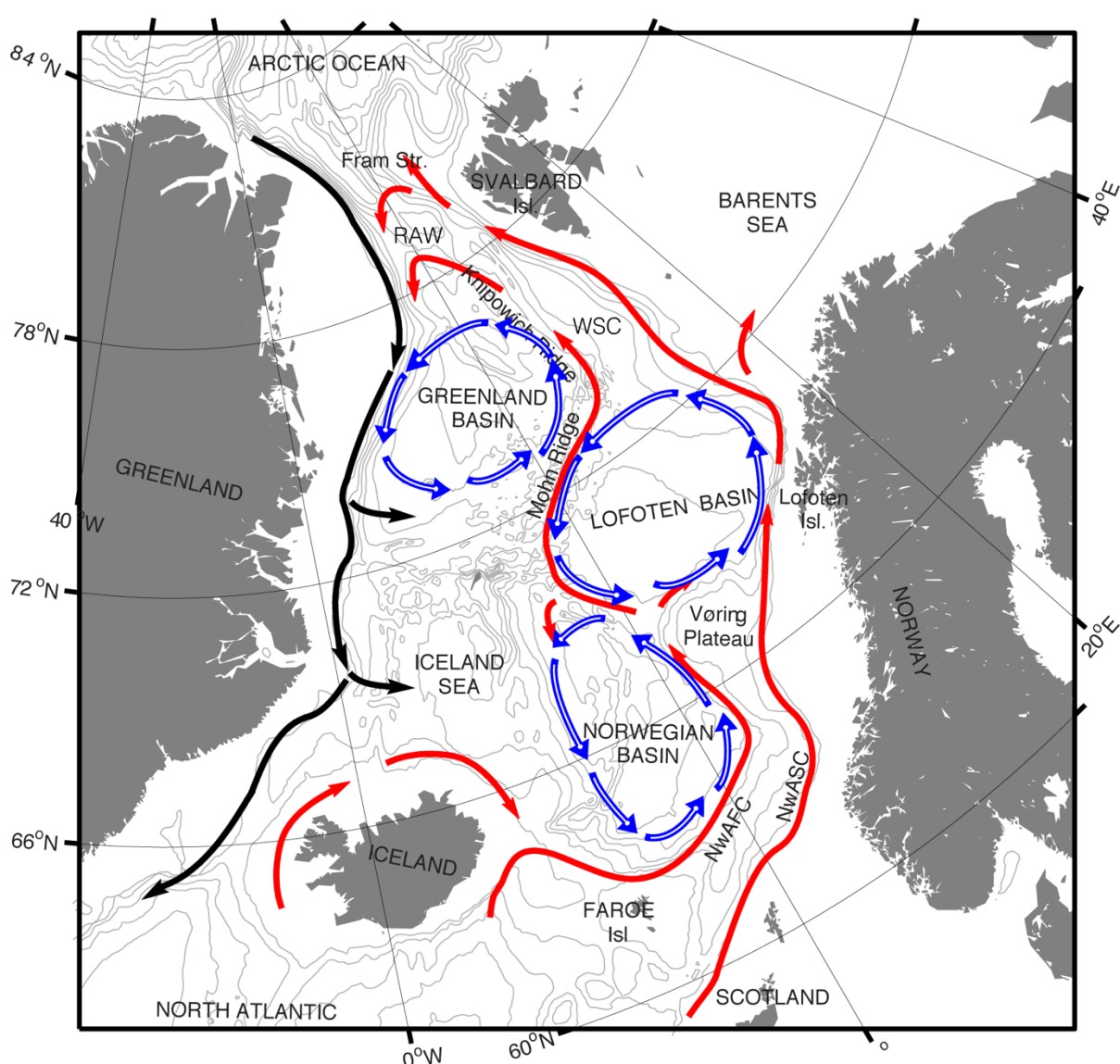

**Figure 1: The Nordic Seas with schematic water pathways showing the northward flowing Atlantic Water in the surface (red) and southward flowing East Greenland Current (black). The two branches of the Norwegian Atlantic Current, the Norwegian Atlantic slope current (NwASC) and Norwegian Atlantic front current (NwAFC) are represented by red arrows. The cyclonic gyre circulations in the Norwegian Basin, Lofoten Basin and Greenland Basin are indicated in blue. See Chatterjee et al. (2018) and Raj et al. (2015) for details. Grey isobaths are drawn for every 600 m. The location of the Arctic Front in the Norwegian Sea coincides with the location of the NwAFC.**

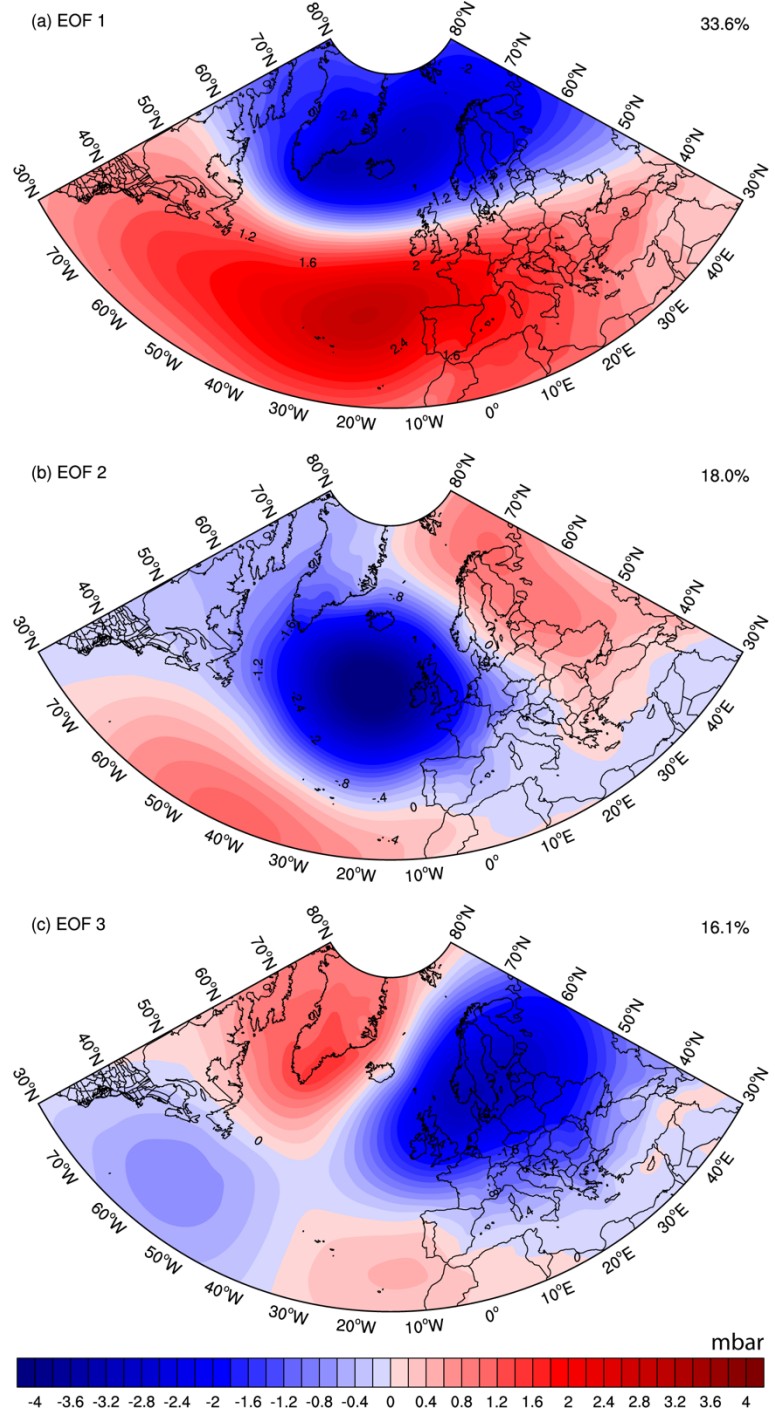

**Figure 2: First three EOF modes of deseasoned and detrended ERA Interim MSLP (1991-2015) multiplied by the standard deviation of the corresponding principle components. The number in right corner of each panel indicates the percentage of variance explained.**

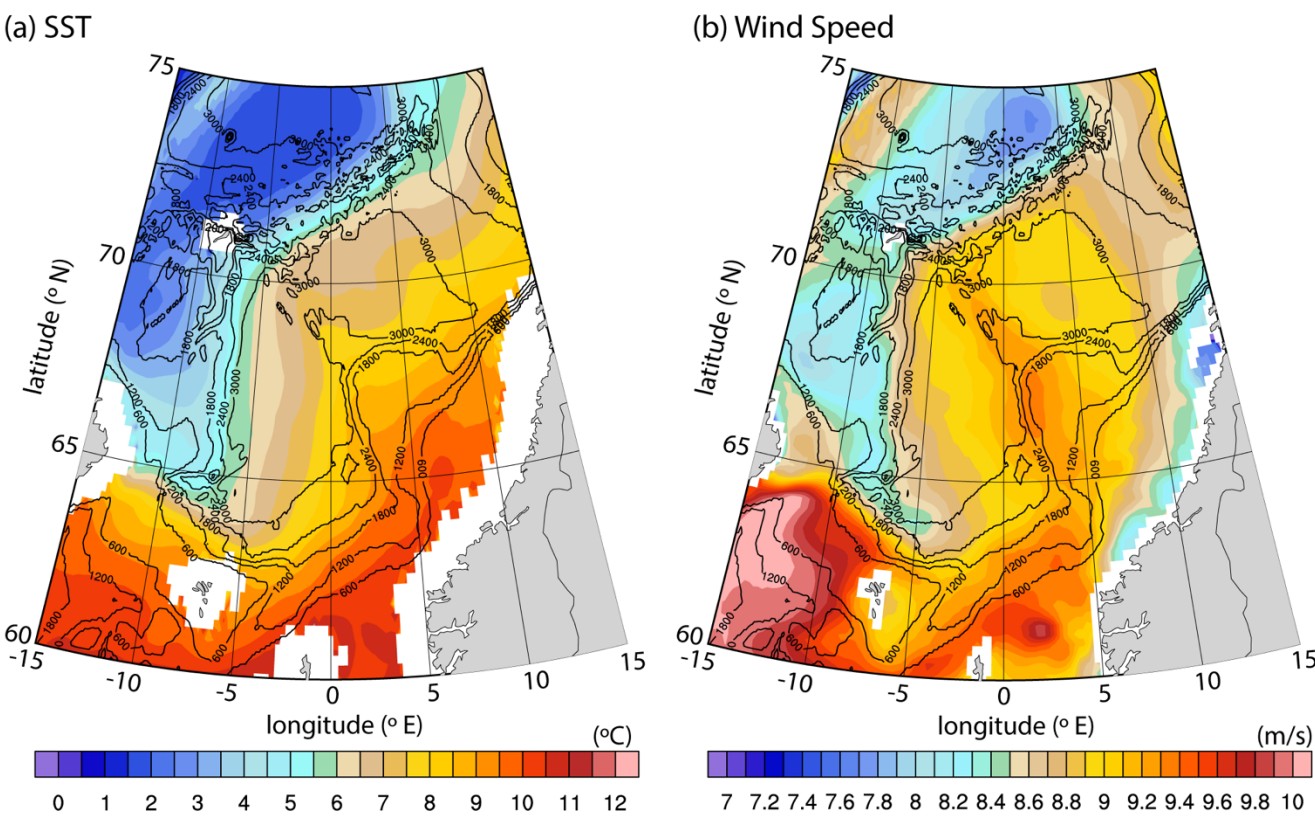

**Figure 3: Climatological mean (2002-2009) (a) AMSRE SST (ºC) and (b) QuikSCat wind speed (m/s). Black isobaths are drawn for every 600 m.**

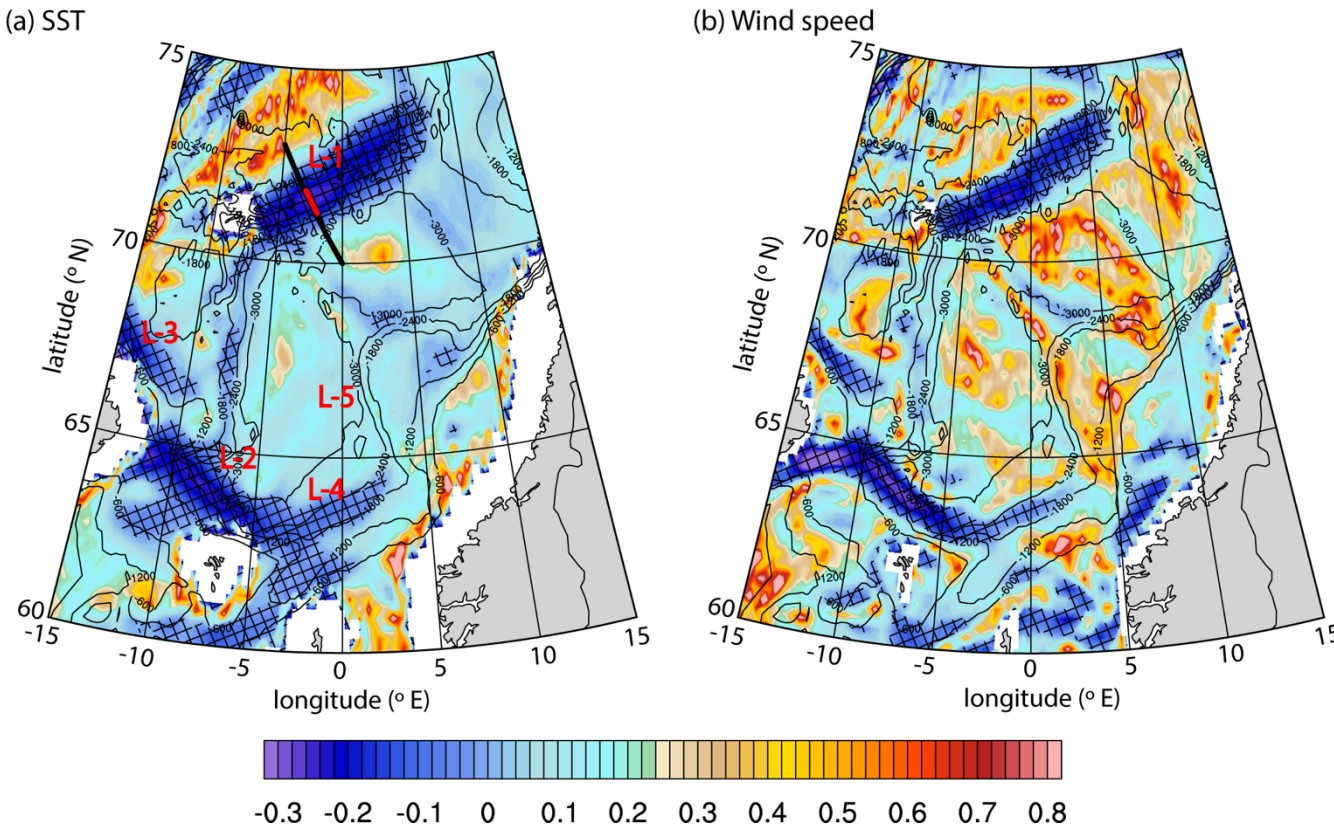

**Figure 4: Singularity exponents estimated from the climatological mean (2002-2009) (a) AMSRE SST and (b) QuikSCat wind Speed. Black isobaths are drawn for every 600 m. In panel a, the bold black line shows the location of the vertical section across the Mohn Ridge shown in Figures 7, 8 and 10. The position of the core of the AF shown in Fig. 7 is marked as red over the bold black line. Locations 1, 2, 3,4 and 5 are represented respectively by L-1, L-2, L-3, L-4 and L-5. Negative values are highlighted (black stripes).**

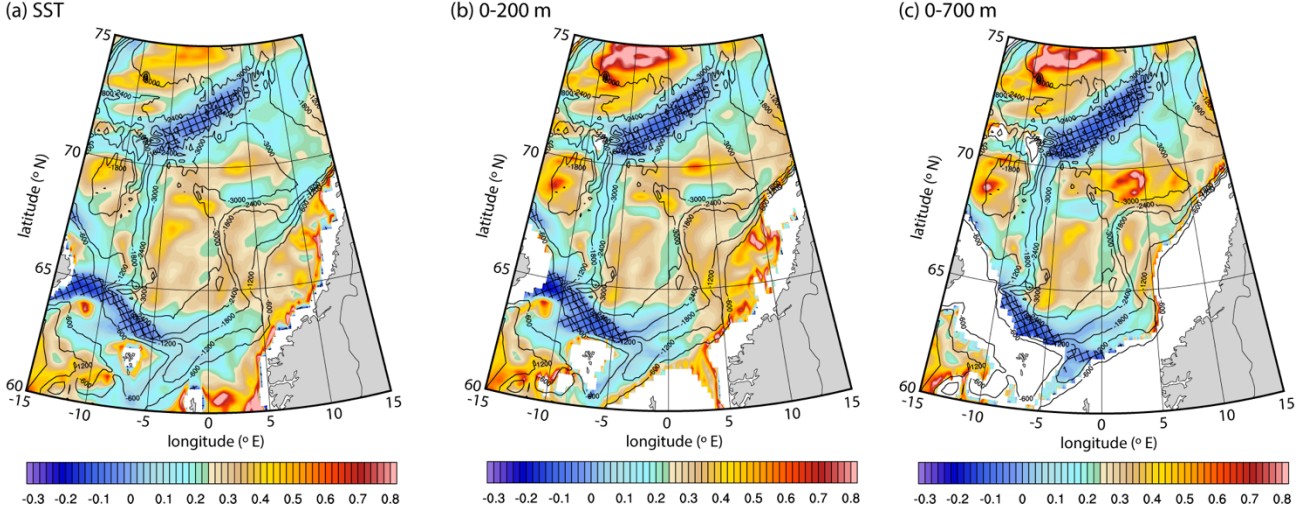

**Figure 5: Singularity exponents estimated from TOPAZ (a) SST, (b) upper 200 m and (c) upper 700 m depth averaged potential temperature for the time period 1991-2015. Black isobaths are drawn for every 600 m. Regions shallower than 200 m and 700 m are masked respectively in panels b and c. Negative values are highlighted (black stripes).**

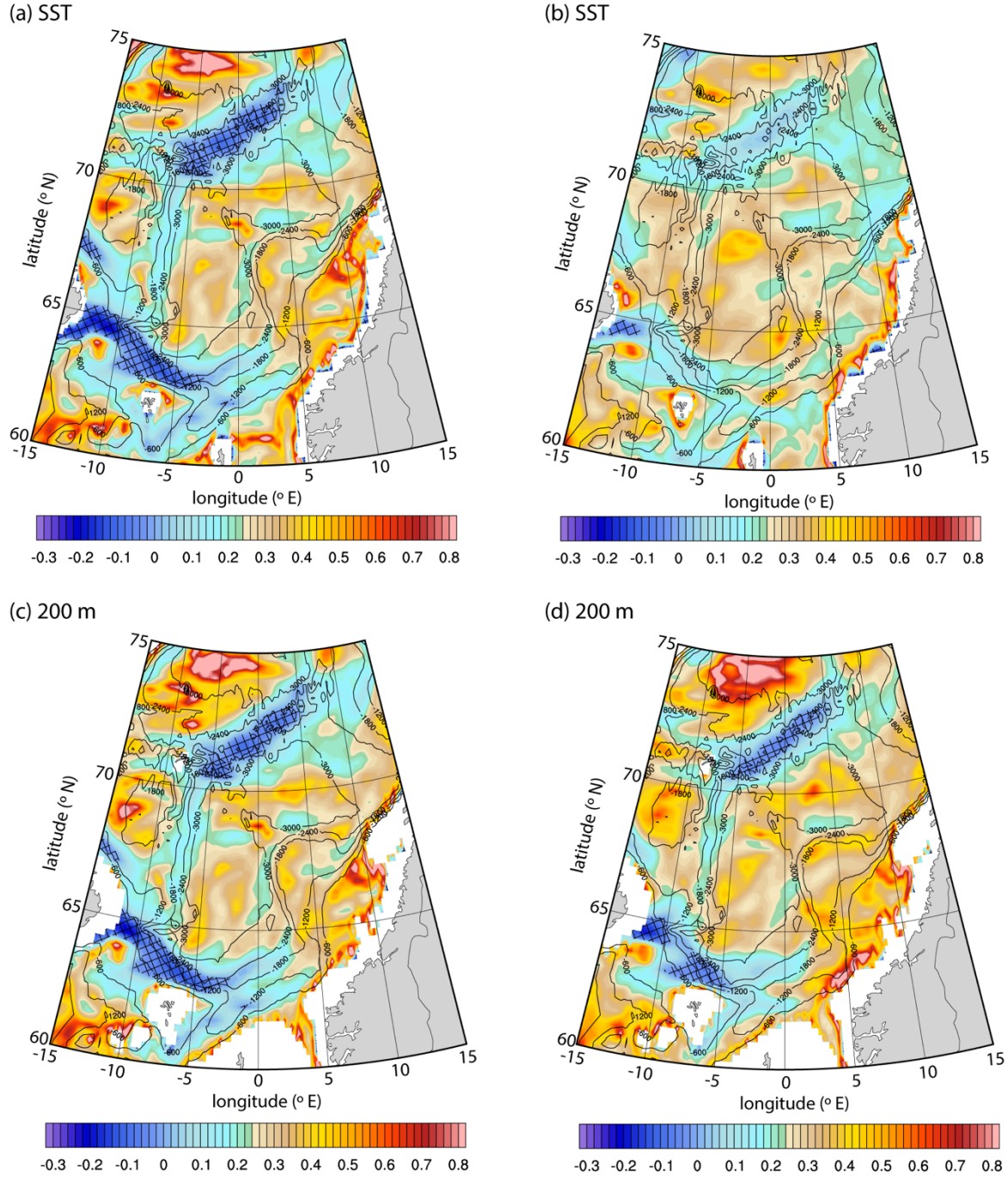

**Figure 6: Singularity exponents estimated from TOPAZ (a-b) SST and (c-d) 200 m potential temperature for (a, c) winter (DJF) and (b, d) summer (JJA) for the time period 1991-2015. Negative values are highlighted (black stripes).**

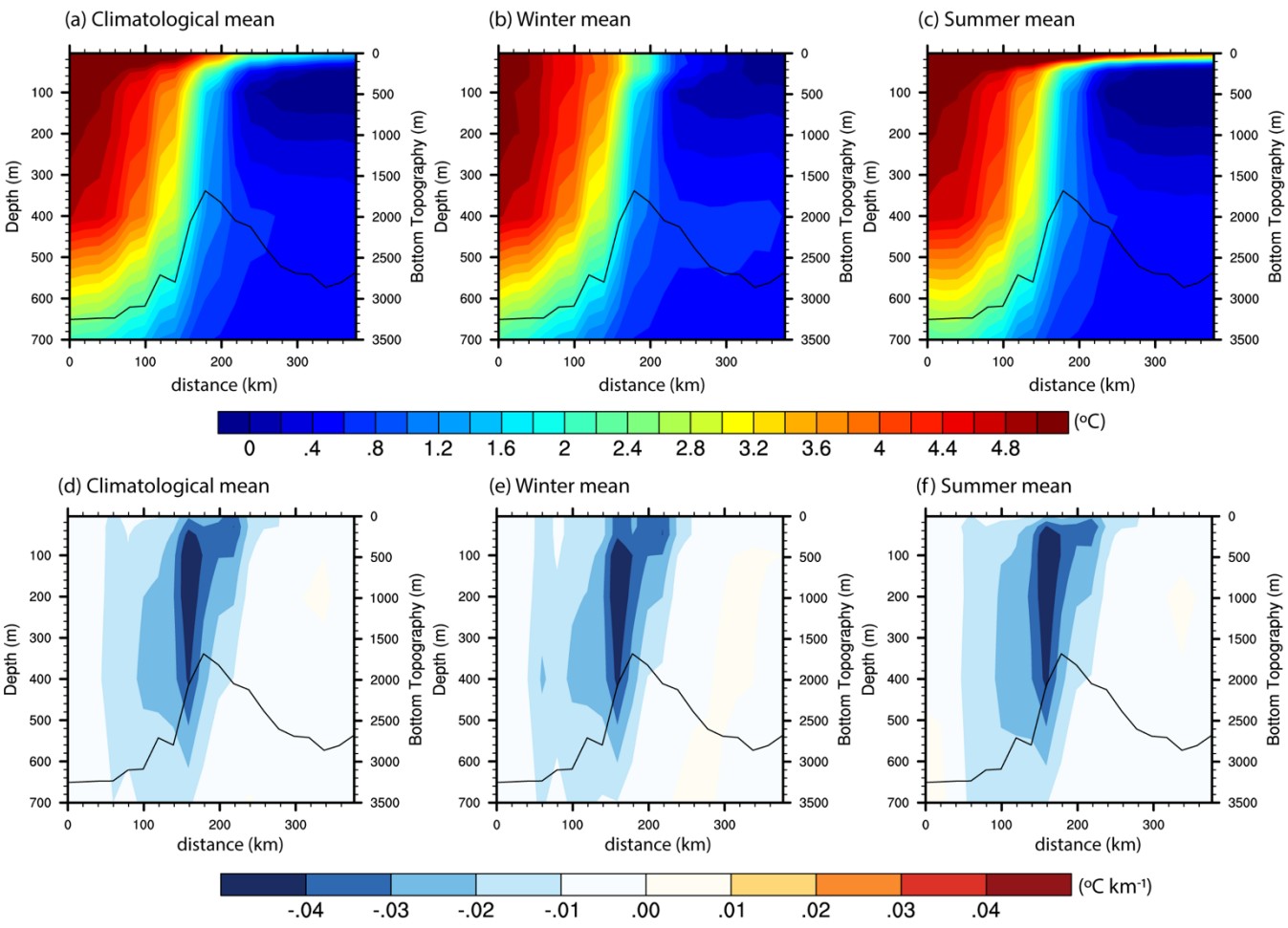

**Figure 7: Temperature (a-c) and its horizontal gradient (d-f) along the hydrographic section across the Mohn Ridge. Climatological mean (a, d), winter (b, e) and summer (c, f). The bottom topography along the section is indicated by the black contour.**

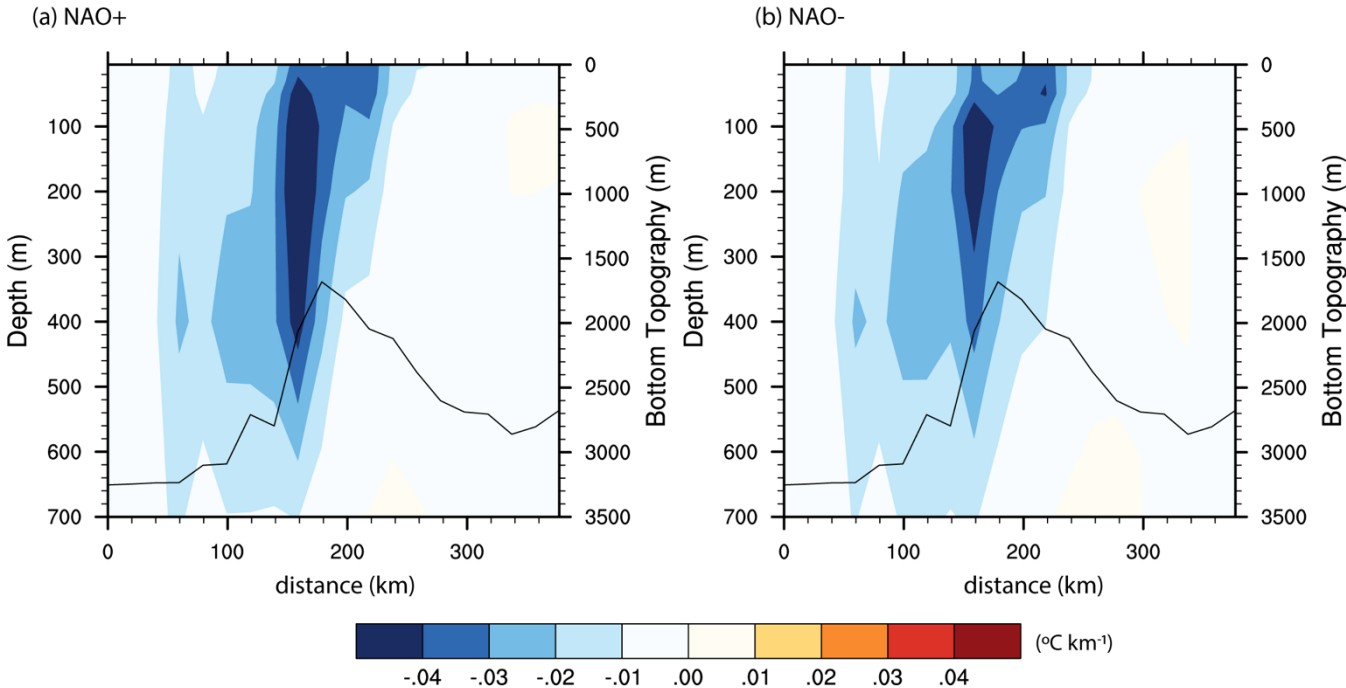

**Figure 8: Gradient in the mean potential temperature (ºC /km) along the hydrographic section across the Mohn Ridge during (a) NAO+ and (b) NAO−. The bottom topography along the section is indicated by the black contour.**

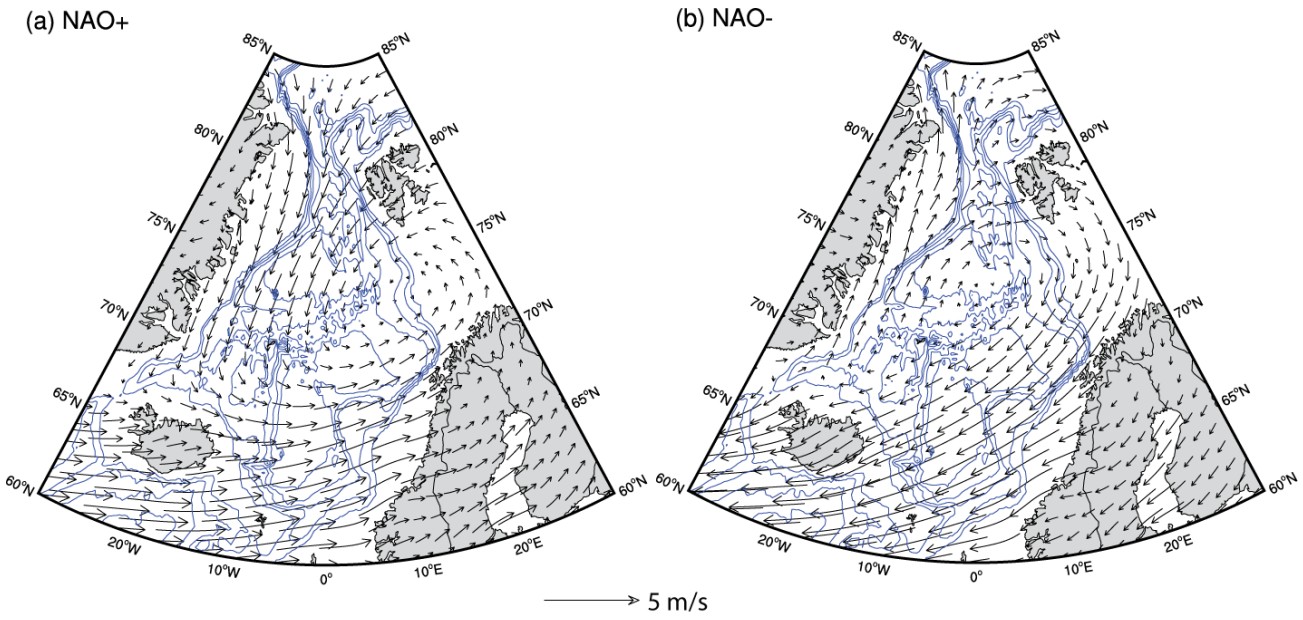

**Figure 9: Composite map of anomalous winds (vectors), during (a) NAO+ and (b) NAO−. Anomalies are computed using monthly climatology fields for the period 1991-2015. Blue isobaths are drawn for every 600 m. Only significant anomalies (10% significance level) are shown.**

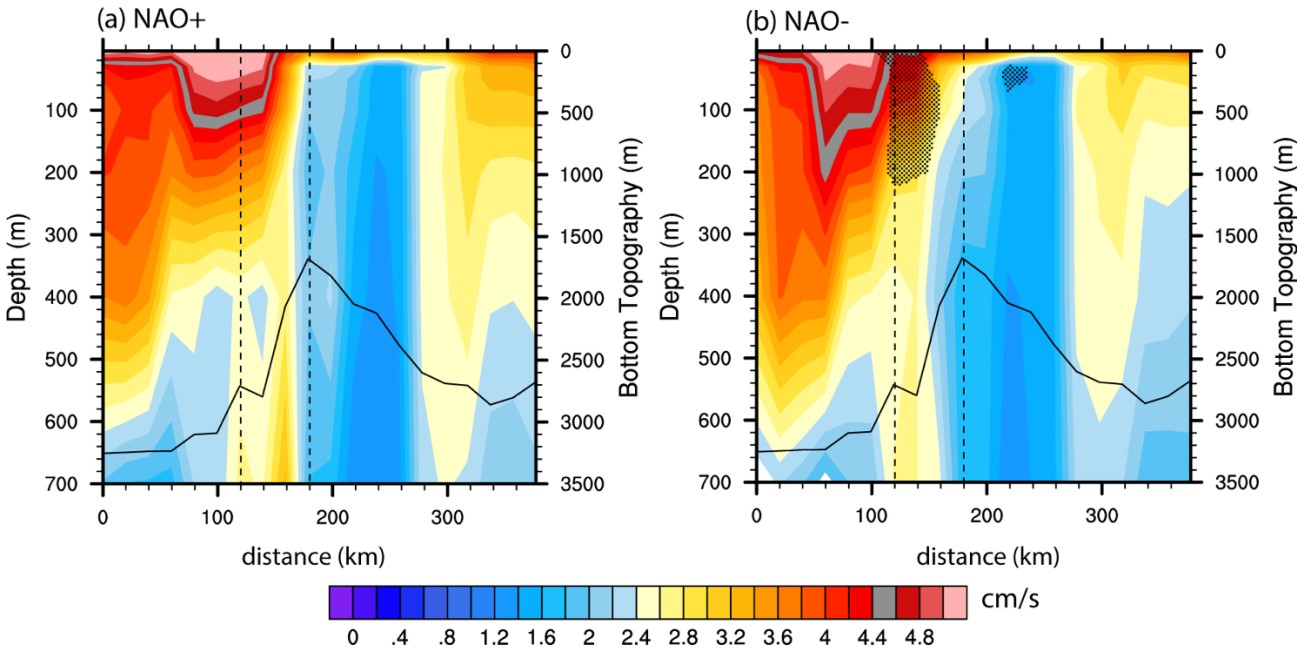

**Figure 10: Composite of currents speed (cm/s) along the hydrographic section across the Mohn Ridge during (a) NAO+ and (b) NAO−. The dashed vertical lines show the position of the core of the AF. Black stripped region in panel b indicates the region where the composite mean speed is significantly different from the climatological mean speed at 10% significance level.**

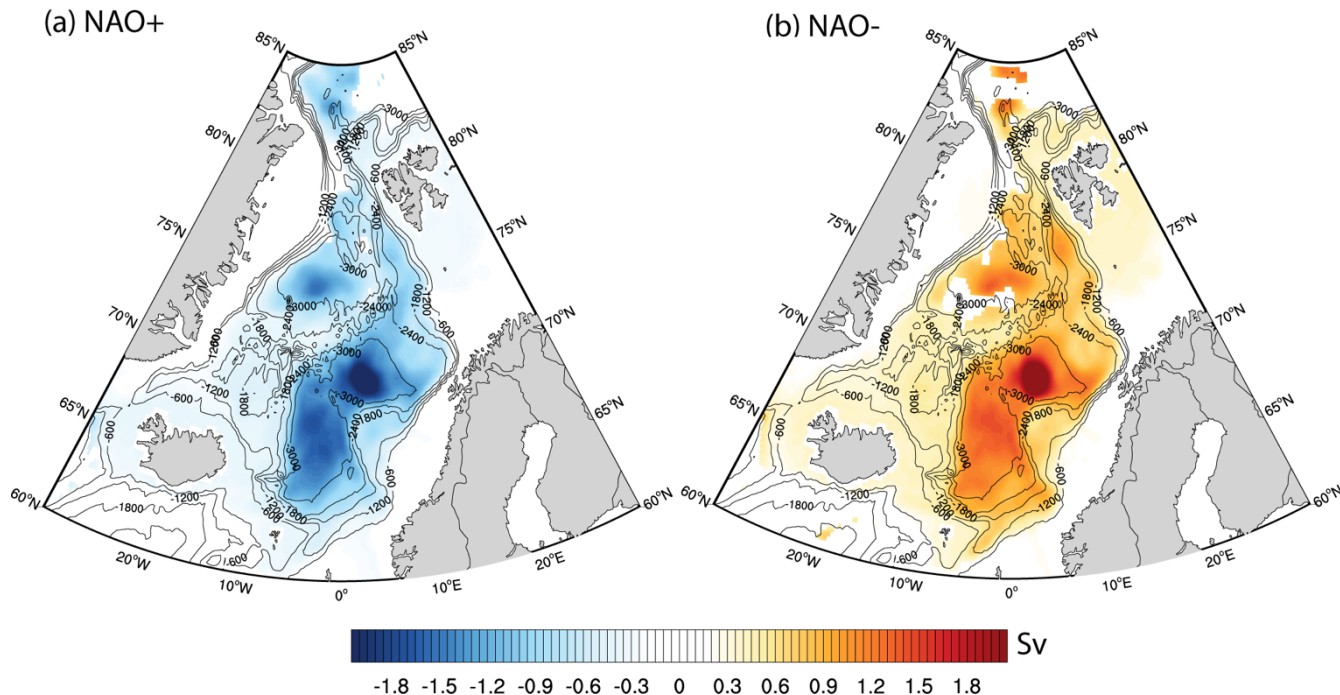

**Figure 11: Composite map of anomalous BSFD, during (a) NAO+ and (b) NAO−. Black isobaths are drawn for every 600 m. Anomalies are computed using monthly climatology fields for the period 1991-2015. Only significant anomalies (10% significance level) are shown.**

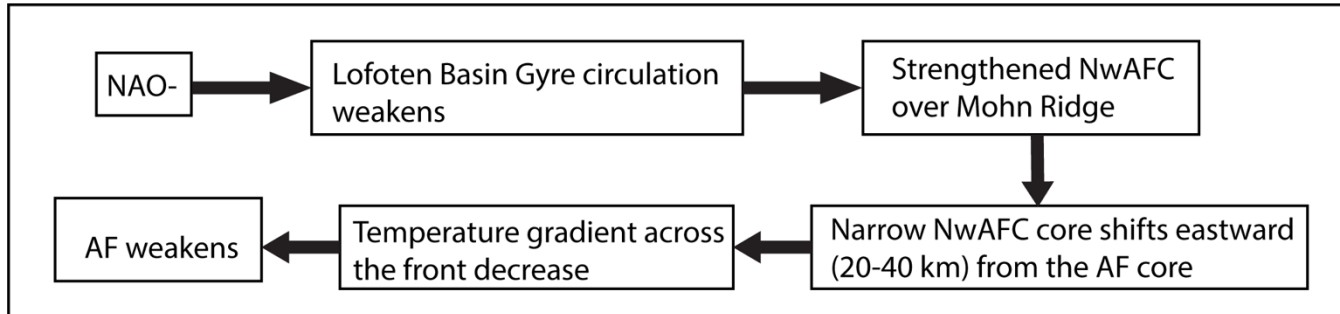

**Figure 12: Graphical representation of the different processes associated with the weakening of the AF during NAO−.**