# Peer review of "The Arctic Front and its variability in the Norwegian Sea"

_Ocean Science, 2018_

## Referee Comment (RC1) · Belonenko (Referee) · 28 Jan 2019

Reviev on the paper "The Arctic Front and its variability in the Norwegian Sea" by Roshin P. Raj, Sourav Chatterjee, Laurent Bertino, Antonio Turiel, Marcos Portebella

The paper in devoted to position clarification of the Arctic front in the Norwegian Sea based on TOPAZ Reanalysis data as well as satellite and atmospheric reanalysis data. Authors apply a new method of front detecting using so-called Singularity analysis. The results obtained are plausible and they are of great interest to specialists. The paper is written in good language and well illustrated. Thus, I suggest that this paper might be published with minor revision.

Below I offer a few comments on the text of the article that can be taken into account

by the authors.

P 2, line 13. "The NwAFC on its way to the north encounters three deep currents (Fig. 1), one over the Mohn Ridge flows in the opposite direction". – Second part of this statement must be confirmed by the appropriate links.

P 2, line 25. It seems "The location of the AF in the Norwegian Sea coincides roughly with the topography along which the NwAFC flows" is not a good phrase for the reader.

I suggest that the item 3.3 Singularity analysis should be described more detailed since this method is very few used and new. It is unclear main differences the Singularity analysis from the method of maxima gradients of scalar parameters to estimate front location. This is optional, but it would be good for the reader of the paper. An online service provided by the Barcelona Expert Center (http://bec.icm.csic.es/CP34GUIWeb/) is a Private Zone and cannot be checked.

Ph.D., Prof. Belonenko Tatyana, St. Petersburg State University, St. Petersburg, Russia

---

## Referee Comment (RC2) · Anonymous Referee #2 · 1 Apr 2019

The study looks into the structure and variability of the Arctic front in the Nordic Seas using satellite SST and wind data as well as an ocean reanalysis. There are some interesting aspects in this paper, but the study seems to touch (briefly) upon a number of questions that ones looses track of the main objective. Because of this, I do not recommend this study for publication in its present form. The main issues are listed below.

Major and general comments

- The singularity analysis is not well defined and is currently only descriptive without any mathematical formulations.

- It is mentioned that a positive (negative) singularity exponent provides information

about regularity (irregularity). This does not add much to the understanding of the method unless you thoroughly explain it. Moreover, Figures 4-6 show exponents less than +0.3 in blue colours, but you discuss most of these results as they were strictly negative.

- What is the reason for these three atmospheric modes when the North Atlantic Oscillation is apparently the most important mode? Are the other modes and their explained variance even significant?

- The timescale of interest is not well defined; you show most of your results for summer and winter but the atmospheric indices you base your composite analysis on are in fact on monthly time scales. I cannot therefore reconcile the results presented in this study and this is a major issue.

- Consider showing the significant regions for the composite analysis.

- I do not see the difference between the Arctic Front and the Norwegian Atlantic Front Current? You talk about the former and make a sudden transition to the latter. What is the difference over the Mohn Ridge?

- The SST-wind relationship at the fronts in the Nordic Seas is interesting but rather descriptive and not convincing at this stage. This needs more careful investigation and needs to be mathematically formulated.

- Does the ocean reanalysis also assimilate the same satellite data you are using? If so, are the similarities you find surprising?

- Some of the texts in the results do not simply fit in there and should be removed or moved to the introduction (one of the many examples is in pp. 7, line 2-3 about seabirds. Another example is the discussion about mesoscale eddies, which I do not see how it fits in)

- There is quite some speculation in the first paragraph of page 8, which needs to be made more sound, especially in relation to the reduced gravity model of Orvik (2004).

This is an important part of the paper.

---

## Author Comment (AC1) · 29 Apr 2019

We thank Prof. Belonenko for the detailed evaluation of our manuscript and constructive suggestions. All comments will be taken into account during the revision of the manuscript. A point-to-point response to the comments is provided in the supplement file.

Please also note the supplement to this comment:
https://www.ocean-sci-discuss.net/os-2018-159/os-2018-159-AC1-supplement.pdf

---

## Author Response (AR1)

**Response to comments from Referee #1**

Reviev on the paper "The Arctic Front and its variability in the Norwegian Sea" by Roshin P. Raj, Sourav Chatterjee, Laurent Bertino, Antonio Turiel, Marcos Portebella. The paper in devoted to position clarification of the Arctic front in the Norwegian Sea based on TOPAZ Reanalysis data as well as satellite and atmospheric reanalysis data. Authors apply a new method of front detecting using so-called Singularity analysis. The results obtained are plausible and they are of great interest to specialists. The paper is written in good language and well-illustrated. Thus, I suggest that this paper might be published with minor revision.

Below I offer a few comments on the text of the article that can be taken into account by the authors.

We thank Prof. Belonenko for her constructive comments. A point-to-point response is given below. The revised text in the manuscript is highlighted in blue.

P 2, line 13. "The NwAFC on its way to the north encounters three deep currents (Fig. 1), one over the Mohn Ridge flows in the opposite direction". – Second part of this statement must be confirmed by the appropriate links.

We agree. This statement is now supported by a reference (Orvik, 2004). Page 2, line 18.

P 2, line 25. It seems "The location of the AF in the Norwegian Sea coincides roughly with the topography along which the NwAFC flows" is not a good phrase for the reader.

We agree. The text is be redrafted. Page 2, line 24-25.

I suggest that the item 3.3 Singularity analysis should be described more detailed since this method is very few used and new. It is unclear main differences the Singularity analysis from the method of maxima gradients of scalar parameters to estimate front location. This is optional, but it would be good for the reader of the paper. An online service provided by the Barcelona Expert Center (http://bec.icm.csic.es/CP34GUIWeb/) is a Private Zone and cannot be checked.

Yes, we agree. In the revised version of the manuscript we provide descriptive details of the singularity analysis in the Appendix Section (pp.10, line 5-31; pp.11, line 1-18).

Notice that the main difference between maxima gradients and singularity exponents is that singularity exponents are normalized so the absolute value of the gradient is irrelevant, what is important is the degree of correlation between nearby gradients (and singularity exponents are dimensionless measures of that correlation). Regarding the Singularity Analysis Service, it is only private in the sense that only registered users can access the service, but registration is free.

*Orvik, K. A., 2004. The deepening of the Atlantic water in the Lofoten Basin of the Norwegian Sea, demonstrated by using an active reduced gravity model. Geophys. Res. Lett. 31, L01306, doi:10.1029/2003GL018687.*

**Response to comments from Anonymous Referee #2**

The study looks into the structure and variability of the Arctic front in the Nordic Seas using satellite SST and wind data as well as an ocean reanalysis. There are some interesting aspects in this paper, but the study seems to touch (briefly) upon a number of questions that ones looses track of the main objective. Because of this, I do not recommend this study for publication in its present form. The main issues are listed below.

We thank the reviewer for the constructive comments. Below we provide point-to-point response of the major and general comments. The revised text in the manuscript is highlighted in blue.

Major and general comments

- The singularity analysis is not well defined and is currently only descriptive without any mathematical formulations.

We agree. In the revised version of the manuscript we provide descriptive details of the singularity analysis in the Appendix Section (pp.10, line 5-31; pp.11, line 1-18).

- It is mentioned that a positive (negative) singularity exponent provides information about regularity (irregularity). This does not add much to the understanding of the method unless you thoroughly explain it. Moreover, Figures 4-6 show exponents less than +0.3 in blue colours, but you discuss most of these results as they were strictly negative.

We thank the reviewer for pointing out this. In the revised version of the manuscript, the methodology is explained thoroughly in the Appendix Section (pp.10, line 5-31; pp.11, line 1-18).
In Figures 4-6, negative singularity exponents are highlighted (stripped) in order to distinguish it from positive values. Kindly note that there are no major changes in the discussion based on the updated figures.

- What is the reason for these three atmospheric modes when the North Atlantic Oscillation is apparently the most important mode? Are the other modes and their explained variance even significant?

It is true that the North Atlantic Oscillation (NAO) is the most dominant atmospheric mode in the North Atlantic and Nordic Seas. However, it is also known that the location of the centers of the NAO dipole can be affected through the interplay with the East Atlantic (EAP) and the Scandinavian (SCAN) teleconnection patterns (Moore et al., 2012; Chafik et al., 2017). Also note that even though the impact of NAO on the slope current is well-known, its impact on the front current and on the Arctic-Front is not very clear.

Also note that recent studies show that even though EAP and SCAN only explains less than 20% (EAP: 18% and SCAN: 16%) of the MSLP variability, its effect on the Nordic Seas cannot be neglected (Comas-Bru and McDermott, 2013; Chafik et al., 2017).

We agree that reasoning needs to be mentioned in the manuscript. We thank the reviewer for pointing this out. The updated version of the manuscript includes the above-mentioned points (pp. 2, line 33; pp.3 line 1-6; pp.4, line 4-6).

- The timescale of interest is not well defined; you show most of your results for summer and winter but the atmospheric indices you base your composite analysis on are in fact on monthly time scales. I cannot therefore reconcile the results presented in this study and this is a major issue.

We agree that there is confusion in the use of different timescales. The first part of the paper is devoted to the seasonal variation of the Arctic Front. Whereas the composite analysis is meant to show the variation related with different climate modes, for which the monthly time scales are the most relevant. We believe that these are two separate aspects and should not be mixed. This has been clarified in the revised version of the manuscript (pp.2, line 28-30; pp.3, line 4-6; pp.8, line 2). We thank the reviewer for pointing this out.

- Consider showing the significant regions for the composite analysis.

We have included significance in composite analysis (Figures 9, 10 and 11).

- I do not see the difference between the Arctic Front and the Norwegian Atlantic Front Current? You talk about the former and make a sudden transition to the latter. What is the difference over the Mohn Ridge?

We agree that the reviewer has a valid point, since the Norwegian Atlantic Front Current (NwAFC) is a baroclinic current. However, we also have reasons for considering them separate over the Mohn Ridge. In our study the Arctic Front is defined as the region with the maximum in temperature gradient, one of the classical methods used to distinguish between two distinct water masses, in our case the Atlantic Water and the Arctic Waters. Moreover, our study shows that the core of the NwAFC is stronger during -NAO, while the AF is weaker during the same period. Our results show that the shift (spreading and narrowing) in the core (defined by the maximum speed) of the NwAFC associated with the variability in the deep ocean circulation forced by atmospheric variability, which is then linked to the variability in the strength of the Arctic Front (Figure 12). Hence they cannot be considered the same.

In the updated version of the manuscript the text is revised where the sudden transition noted by the reviewer is avoided (pp.8, line 5-8, 17-24).

- The SST-wind relationship at the fronts in the Nordic Seas is interesting but rather descriptive and not convincing at this stage. This needs more careful investigation and needs to be mathematically formulated.

We agree that more quantitative work needs to be done in terms of SST and wind interaction. However, we believe that the qualitative agreement supports well enough the main goal of the manuscript (variability of the Arctic Front), and the SST-wind topic in itself has the potential to be a separate paper. We think that it should still be of interest to the readers to see the new results. In the revised version we will make sure that these results are not influencing the major conclusions of the study and in the discussion part we will also highlight the need for a detailed analysis of the wind-SST relationship over the Arctic Front. We believe that the inclusion of these results in this study will result in new studies focusing exclusively on the subject (pp. 5, line 27-29).

- Does the ocean reanalysis also assimilate the same satellite data you are using? If so, are the similarities you find surprising?

Yes, TOPAZ reanalysis assimilates measurements including along-track altimetry data, sea surface temperatures, sea ice concentrations and sea ice drift from satellites along with in-situ temperature and salinity profiles. Hence, we agree that the similarities are not surprising, although not warranted due to the residual errors of assimilation. But the intent is to show that the reanalysis data is able to reproduce the Arctic Front. This is clearly mentioned in the revised version of the manuscript (pp. 6, line 30-33).

- Some of the texts in the results do not simply fit in there and should be removed or moved to the introduction (one of the many examples is in pp. 7, line 2-3 about seabirds. Another example is the discussion about mesoscale eddies, which I do not see how it fits in)

We agree. In the revised version, the discussion about the seabirds is limited to the introduction part.

However, note that the discussion on the mesoscale eddies is needed to explain the absence of the Arctic Front signature in the singularity exponent map over the northern Vøring Plateau (pp.6, line 8-16). Please note that the discussion on the mesoscale eddies is supported by the supplementary Figure S2.

- There is quite some speculation in the first paragraph of page 8, which needs to be made more sound, especially in relation to the reduced gravity model of Orvik (2004). This is an important part of the paper.

We thank the reviewer for his comment. We have revised the paragraph (pp.8, line 25-33, pp.9, line 1-8).

[revised manuscript text omitted]

---

## Author Response (AR2)

**Response to comments from Referee #3**

The authors study the variations in the position and intensity (in terms of horizontal temperature gradient) of the Arctic Front in the Norwegian Sea, on the basis of surface SST and surface winds remote sensed data as well as TOPAZ model outputs.
The analysis is conducted on a climatological scale, and explore the seasonal variability and the influence of atmospheric variability through composite analyses.

The manuscript is well structured, and clearly written. The mechanistic description proposed to link the influence of atmospheric variability and the morphology of the AF front at the Mohn Ridge is of great interest.
I therefore suggest to proceed with the publication of this manuscript, acknowledging some modifications, and suggestions listed below :

We thank the reviewer for going through the manuscript and for the constructive comments. Below we provide a point-by-point response to the major and minor comments. We have revised our manuscript accordingly. The revised text in the manuscript is highlighted in blue.

Major Comments :
1 ) In agreement, with previous reviewers I think that the singularity exponent approach should be better introduced. While it is good to retain technical details in the appendix, Sect 3.3 should better explain in simple terms:

In the revised version of the manuscript, the Singularity analysis is better introduced and explained in simple terms by also addressing the comment below. See page 4, lines 19-30.

** Why is this approach favored to simple horizontal gradients ? What does it brings in addition (for instance the answer to Referee #1 can be used in this regard). The reason to use this level of complexity is not evident and should be better justified.

There are several methods in practice to identify ocean fronts from satellite data which have a higher complexity than the simple horizontal gradient method (e.g., Cayula and Cornillon,1992,1995; Garcia-Olivares et al., 2007; Turiel et al., 2008). One of them is the singularity analysis, which is a Eulerian method exploiting the scaling properties of the spatial correlations of the gradients of a given scalar field. Mallat and Huang (1992) introduced the Singularity analysis of scalar variables in the context of wavelet analysis. The singularity analysis aims to obtain a dimensionless measure known as the singularity exponent at each point which represents the degree of irregularity at that location. Note that the singularity exponent is a continuous extension of classical concepts such as continuity or differentiability.

The main difference between maxima gradient method (another widely used methodology) and singularity exponents is that singularity exponents are normalized such that the absolute value of the gradient is irrelevant, while the degree of correlation between nearby gradients is exploited (and singularity exponents are dimensionless measures of that correlation). Hence the results from the singularity analysis of different scalars variables (for e.g., SST and windspeed) can be directly compared. Note that Singularity analysis in our study is used to identify the prominent locations of the AF in the Norwegian Sea from satellite SST and wind speed data together with TOPAZ reanalysis. Another widely used methodology in ocean fronts studies is the use of Lyapunov exponents. A main advantage of the Singularity analysis is that it does not require the velocity field to be known and unlike the Lyapunov exponents (Garcia-Olivares et al., 2007), singularity exponents are dimensionless and can be derived from any

scalar quantity, e.g., SST (Turiel et al., 2009) and wind components (Portabella et al., 2012; Lin et al., 2014). Hence even though slightly complex, the Singularity analysis is one of the most effective methods currently available to study the AF in the Norwegian Sea. This is now clarified in revised version of the manuscript (see page 4, lines 19-31).

** Why are the SE used for the horizontal, while the more traditional gradient approach is still favored for the vertical. Are there technical limitation to use SE for the vertical panels? If gradients are suited and sufficient for the vertical discussion, why aren't they for the horizontal ?

The singularity analysis in the first part of the paper is used to identify the prominent locations of the AF in the region from satellite SST and wind speed data together with TOPAZ reanalysis. The results from the singularity analysis of different scalars variables (for e.g., SST and windspeed) can be directly compared since singularity exponents are normalized so the absolute value of the gradient is irrelevant, what is important is the degree of correlation between nearby gradients (and singularity exponents are dimensionless measures of that correlation). The vertical structure of the front is then analyzed using a simple horizontal gradient method over the Mohn Ridge, a location where the front is found to be most prominent by the singularity analysis. The main reason for using a simple horizontal gradient method to estimate the strength of the front is to compare our results with previous studies focusing on ocean fronts (e.g., Piechura and Walczowski, 1995; Lobb et al., 2003). We found that the potential temperature gradient across the core of the AF is comparable to those reported at other high-latitude frontal regions (Lobb et al., 2003). Also, our results are consistent with those reported earlier by Piechura and Walczowski (1995) from a similar section over the Mohn Ridge. The choice of a simple vertical gradient method does not relate to any technical aspect as there are no technical limitations to use Singularity analysis in the vertical.

We agree with the reviewer that the current version of the manuscript lacks clarity on the use of the two methods. We thank the reviewer for pointing this out. The updated version of the manuscript includes the above-mentioned points, thus clarifying the choice of the two methods (see page 4, lines 19-31; page 7, lines 27-29).

2) While the first section (climato and seasonal) address a large area, the second part seems restricted to the Mohn ridge, while there is seemingly enough material and tools to have a larger overview of the depicted mechanism. For instance, Fig9. depicts a substantial anomaly of circulation over the L-2 and L4 locations (Fig 4.) according to NAO phases. Is there any reason to restrict the detail insights to the Mohn Ridge ? Would the changes in circulation (Fig. 11) also affect the AF morphology in other areas evidenced in the first sections (eg. L2; L4) but not discussed in the second part ? Some justification should be given to explain this narrowing of the analysis from the first to the second part of the result section, unless the autors choose instead to extent spatially the discussion of the second part. In particular, Fig 12. describe te mechhanism in general terms, although this mechanism is discussed in details only for the Mohn Rigde.

The first part of the paper, which focuses on the AF of the Norwegian Sea, identifies the front to be most prominent at two locations: the L1 (Mohn Ridge) and L2 locations, as shown in Fig 4. In the previous version of the manuscript, it is mentioned that the analysis focusing on the vertical structure of the front is limited to Mohn Ridge because of the strong signature of the

front found at this location. Even though a reason for limiting our analysis to Mohn Ridge was mentioned in the manuscript, we agree with the reviewer that this justification is not enough to limit the analysis to Mohn Ridge, especially when anomalous winds are found over L2 and L4 during the opposite phases of NAO.

In the revised version of the manuscript (see page 7, lines 29-34; Page 8, line 1; Page 10, lines 1-7) additional text based on the following points is added in order to fully justify our selection:

(1) The circulation as mentioned in the introduction (see Fig 1) is somehow complex in the Mohn Ridge region compared to other regions, e.g., the gyre circulation of the Lofoten Basin is opposite to that of the northward flow of the NwAFC over Mohn Ridge. On the contrary, the flow of the NwAFC at locations L2 and L4 follows the direction of the gyres in that region. Thus, the mechanism described in Fig.12 only applies to the variability of the AF over Mohn Ridge and not over other regions. Therefore, the impact of atmospheric forcing on L2 location needs to be analyzed in detail, although this is out of the scope of this study. In the revised version, the need for a detailed analysis at L2 is highlighted since it is one of the two locations where AF is found to most prominent. This analysis is now recommended as future work in the concluding section;

(2) The variability of the AF over Mohn Ridge is also important due to its proximity to the Jan Mayen Island, which is well-known to be an important breeding region inhabited by large colonies of seabirds. Although higher up in the food chain, the variability of the AF may have an influence on the birds through the impact on biology and fisheries of the region. This is also a reason for selecting Mohn Ridge to analyze the impact of atmospheric forcing on the AF.

Minor Comments :

P4.L27 "projected on a given wavelet" : rephrase, this sounds very vague.

The sentence is redrafted in the revised version of the manuscript (page 5, lines 7-8).

P5l11 : remove 'the' in 'the parts'
Corrected.

P6L6 : 'north' -> 'northeast' ?
Corrected (Page 6, line 17).

P8L30-32 : 'hinderance' or 'hindrance' ?
Corrected (Page 9, line 16).

P9L13: I would refer to Figure 8 when mentioning the weakening of the front.
Added the reference of Figure 8 (Page 9, line 32).

p15 the year for bib entry Umbert et al differs in the text and in the bibliography.
Corrected (Page 4, line 19).

Fig 10. current speed ALONG or ACROSS the section ? Is that the absolute norm of this current component or is it simply going in the same direction all over the panel ? Isn't possible, if using the across-section current component (not the absolute value) to illustrate bot the

NwAFC and the Lofoten basin cyclonic circulation ? This would help in describing the mechanism detailed in Fig 12.

We agree that the reviewer has a valid point in suggesting cross-section velocities in the figure. However, note that although the vertical section is aligned almost perpendicular to the AF, the maximum velocity component of the NwAFC and the gyre circulation are not necessarily perpendicular to the section, and hence cross-section velocities may not give a correct representation of the currents. Instead, the speed along the section can give a better representation of the currents, especially since the direction of the mean currents does not depend on atmospheric forcing. Also, the main focus here is on the lateral shift of the NwAFC along the section during the positive and negative NAO. Figure 10 shows the lateral shift of the NwAFC during negative NAO. Furthermore, while Fig. 10 shows a prominent shift in the location of the NwAFC, Fig 11, shows the change in the Lofoten Basin cyclonic circulation during NAO-, thereby supporting the physical mechanism explained in Fig 12.

The reason for using speeds instead of cross section velocity component is added to the text. (see page 9, lines 1-5)

References

[revised manuscript text omitted]